# Exploring the Interplay between Polyphenols and Lysyl Oxidase Enzymes for Maintaining Extracellular Matrix Homeostasis

**DOI:** 10.3390/ijms241310985

**Published:** 2023-07-01

**Authors:** Carolina Añazco, Janin Riedelsberger, Lorenzo Vega-Montoto, Armando Rojas

**Affiliations:** 1Laboratorio de Bioquímica Nutricional, Escuela de Nutrición y Dietética, Carrera de Nutrición y Dietética, Facultad de Ciencias para el Cuidado de la Salud, Universidad San Sebastián, General Lagos #1190, Valdivia 5110773, Chile; 2Centro de Bioinformática, Simulación y Modelado (CBSM), Facultad de Ingeniería, Universidad de Talca, 1 Poniente 1141, Talca 3462227, Chile; jriedelsberger@utalca.cl; 3Chemical and Radiation Measurement, Idaho National Laboratory (INL), 1705 N. Yellowstone Hwy, Idaho Falls, ID 83415, USA; lorenzo.vegamontoto@inl.gov; 4Biomedical Research Laboratories, Medicine Faculty, Catholic University of Maule, Talca 3480112, Chile; arojasr@ucm.cl

**Keywords:** collagen cross-linking, extracellular matrix, fibrosis, glycation, lysyl oxidase, polyphenols

## Abstract

Collagen, the most abundant structural protein found in mammals, plays a vital role as a constituent of the extracellular matrix (ECM) that surrounds cells. Collagen fibrils are strengthened through the formation of covalent cross-links, which involve complex enzymatic and non-enzymatic reactions. Lysyl oxidase (LOX) is responsible for catalyzing the oxidative deamination of lysine and hydroxylysine residues, resulting in the production of aldehydes, allysine, and hydroxyallysine. These intermediates undergo spontaneous condensation reactions, leading to the formation of immature cross-links, which are the initial step in the development of mature covalent cross-links. Additionally, non-enzymatic glycation contributes to the formation of abnormal cross-linking in collagen fibrils. During glycation, specific lysine and arginine residues in collagen are modified by reducing sugars, leading to the creation of Advanced Glycation End-products (AGEs). These AGEs have been associated with changes in the mechanical properties of collagen fibers. Interestingly, various studies have reported that plant polyphenols possess amine oxidase-like activity and can act as potent inhibitors of protein glycation. This review article focuses on compiling the literature describing polyphenols with amine oxidase-like activity and antiglycation properties. Specifically, we explore the molecular mechanisms by which specific flavonoids impact or protect the normal collagen cross-linking process. Furthermore, we discuss how these dual activities can be harnessed to generate properly cross-linked collagen molecules, thereby promoting the stabilization of highly organized collagen fibrils.

## 1. Introduction

The biosynthesis and maturation of collagen fibrils involves several post-translational modifications including folding, hydroxylation of proline and lysine residues, glycosylation of lysine and hydroxylysine residues, disulphide bonding, trimerization, secretion, processing, self-assembly and cross-linking [1,2,3,4,5,6,7]. The lysyl oxidase family catalyze the final enzyme step in the biosynthetic collagen cross-linking [8,9]. This last reaction is a meticulous sequential process that occurs outside the cell, and is classified as a physiologically controlled enzymatic process, which is essential to the creation of a standard suprafibrillar architecture of collagen fibers that contribute to the mechanical properties and structural organization of tissues [2,7,10,11].

Lysyl oxidase-mediated collagen cross-linking is an extracellular post-translational modification crucial for collagen structure stabilization and the tensile strength properties of connective tissues [5,8,9,12]. LOX catalyzes the oxidative deamination of ε-amine lateral chains of lysine and hydroxylysine in telopeptide regions in growing fibrils, which originate reactive aldehydes (α-aminoadipic-δ-semialdehydes) that can condense each other or lysine and hydroxylysine residues in the helical domain to form inter- and intramolecular divalent cross-links, which spontaneously condense to form a variety of mature and permanent covalent cross-links. Therefore, this final enzymatic reaction is essential to the formation of immature and mature cross-links that stabilize tropocollagen macromolecules during the formation of collagen fibrils, and it is critically important to support normal tissue remodelling and growth [8,9,11,13]. 

In contrast, glycation is a non-enzymatic reaction that can modify collagen cross-linking in fibrils, producing advanced glycation end-products (AGEs), which are associated with the altered mechanical properties of collagen fibers and stiffening of aged tissues [13,14]. Such uncontrolled processes occur when sugar molecules attach covalently to lateral ε-amine chains of lysine residues in collagens. The non-enzymatic collagen cross-linking by glycation is considered to negatively affect the typical functions of connective tissues. Two lysine-arginine cross-links have been identified in collagen: the first is pentosidine, which is a fluorescent product formed from ribose, and the second is glucosepane derived from glucose. The latter is found abundantly in senescent skin collagen [7]. The collagen turnover decreases during normal ageing and the accumulation of AGEs can easily bind and cross-link collagen, modifying the physical properties of fibers, which are deformed, altering the homeostasis between the degradation and synthesis of the extracellular matrix (ECM) [15]. 

On the other hand, polyphenols are a group of phytochemicals found in plants, fruits, vegetables, floral tissues, stems, bark and roots. They are widely known for their general antioxidative capacity [16,17,18]. However, in addition to their antioxidant activity, some polyphenols also have the potential to promote the oxidative deamination of primary amines by oxidation to the corresponding o-quinone derivative [19,20,21,22]. Interestingly, through the formation of the o-quinone derivative bioflavonoids, α-aminoadipic-5-semialdehyde can form in collagen and elastin through the oxidation of the ε-amine group of lysine residues [20,23,24,25,26]. This activity has been related to non-specific chemical catalysis because it is not blocked by β-Aminopropionitrile (BAPN), a competitive inhibitor of the lysyl oxidase family [27]. Notably, polyphenols can also inhibit protein glycation and have been postulated as potential antiglycation agents, mainly due to their antioxidant and chelating activities, which can act as compounds inhibiting or breaking AGE-induced cross-links in collagens [28,29]. However, despite the plentiful findings about polyphenols reported to date, the specific biological functions of such polyphenols on collagen cross-link types at their molecular level are still unclear. The influence of bioflavonoids on the activity of other members of the lysyl oxidase family has also not been explored.

Therefore, the objective of this review is to elucidate the oxidative properties of certain polyphenols in facilitating the formation of both immature and mature cross-links, which contribute to the stabilization of tropocollagen macromolecules. Additionally, this review delves into the inhibitory and disruptive effects of polyphenols on the formation of advanced glycation end-products (AGEs) in collagen. By shedding light on these mechanisms, this research lays the foundation for the development of active compounds capable of modulating oxidative deamination and interfering with the glycation of lysine ε-amino groups within collagen fibrils.

## 2. Fibrillar Type I Collagen

Extracellular matrices provide essential support and create a conducive environment for the assembly of large macromolecular structures, which consist of macromolecules and peptides, including collagen precursors and mature collagen. These structures combine to form larger fibrillar arrangements [30,31]. The presence of collagen within the connective tissues’ fundamental framework is of the utmost importance for establishing a proper tissue architecture. These intricate networks of macromolecules exert specific mechanical forces that profoundly impact cellular behavior and morphology [31]. Collagens are the most abundant proteins in the extracellular matrix (ECM), where most interact with other ECM proteins to form the structural network of tissues [31]. Collagen interaction with cells involves several processes, including cell differentiation, mechanical strength, proliferation, cell adhesion and migration [1,7]. Based on their supramolecular assemblies, collagens are classified into four subfamilies: fibrils, beaded filaments, networks and anchoring fibrils [7,31,32,33]. Twenty-eight types of collagens have been found in mammals to date, with type I collagen being the most common and abundant form in animals [7,34]. Type I collagen, a fibril-forming collagen, is found in several tissues, including skin, bones, tendons, vascular system and cornea [1,7]. The triple helix is a common structural motif to all collagen types, and it stabilizes by the presence of the characteristic repeating amino acid sequence Gly-X-Y, where glycine occupies every third position, and X a Y are often proline and hydroxyproline, respectively [7]. Three parallel α-chains twisted around each other build procollagen molecules with a right-handed triple helix. Type I collagen mainly comprises heterotrimers containing one α2 and two α1 chains (Figure 1) [35]. The triple helix is conformationally stabilized by interchain hydrogen bonds formed by proline and hydroxyproline. Hydroxyproline is formed by prolyl hydroxylase in a reaction that requires essential cofactors such as Fe^2+^, 2-oxoglutarate, O_2_ and vitamin C (Figure 1A) [5]. Other amino acids, including lysine, arginine, glutamate and aspartate, participate in the formation of electrostatic attractions in type I procollagen and provide conformational stability to the triple helix [1,36].

It has been proposed that lysine hydroxylation and lysine glycosylation are crucial for the formation of reducible and mature cross-links produced via the lysyl oxidase system and for stabilizing the supramolecular structures of fibrils-formed collagens [1,7,37,38]. In particular, lysine residues in procollagens are post-translationally modified by lysyl hydroxylases (LH) (Figure 1A). LH1 catalyzes lysine modifications inside the cell by hydroxylating specific peptidyl lysine residues producing hydroxylysine residues in triple-helical regions, whereas LH2 mediates the hydroxylation of lysine residues in terminal telopeptidyl regions [39]. This hydroxylation occurs almost exclusively in the Y positions of the repeating sequence Gly-X-Y in a well-ordered reaction that requires the same essential cofactors used for proline hydroxylation reactions, which leads to the release of a hydroxylated lysyl residue in the procollagen polypeptide, CO_2_, and succinate (Figure 1A) [16]. An extensive review was published by Yamauchi et al., focusing on post-translational lysine modifications [5].

Collagen molecules undergo essential modifications for their proper folding and supramolecular structure formation. These modifications include the hydroxylation of proline and lysine, as well as glycosylation with O-linked glycosides of hydroxylysine residues. Type I collagen, in particular, relies on these posttranslational changes for crucial functional characteristics [39,40]. The hydroxylation of specific lysine residues along the procollagen α-chains is especially significant for subsequent normal collagen cross-linking [38]. Moreover, certain hydroxylysine residues undergo sequential O-linked glycosylation, which plays a role in controlling collagen extracellular fibrillogenesis, covalent intermolecular cross-linking, mineralization, and cell–matrix interaction [5,39,41,42]. Fibrillar type I collagen undergoes O-glycosylation within the endoplasmic reticulum (ER) on the 5-OH group of 5-hydroxylysine. This process involves the addition of a single galactose, resulting in the formation of galactosyl-5-hydroxylysine (monosaccharide). Additionally, glucose can be added to form glucosylgalactosyl-5-hydroxylysine (disaccharide), although it is less abundant [39,42,43]. Studies have demonstrated that the residues α1/2–87 serve as major O-glycosylation sites, contributing to the formation of divalent covalent intermolecular cross-linking that potentially regulates cross-link maturation in type I collagen [42].

Furthermore, it has been observed that the highly conserved N-glycosylation in the C-terminal globular domain is not essential under normal physiological conditions. This is because it is cleaved during the processing of procollagen outside the cell. However, in conditions where proteostasis is disrupted, such as collagenopathies, this N-glycosylation is believed to be crucial for collagen folding and secretion [44].

It is important to note that these posttranslational modifications differ from glycation, which occurs in pathological conditions and leads to the formation of advanced glycation end-products (AGEs) through protein glycoxidation. In summary, O-glycosylation plays a critical role in the posttranslational modifications of hydroxylysine, which is important for the formation of extracellular covalent crosslinking in the extracellular matrix (ECM). On the other hand, the N-glycosylation of asparagine residues is necessary for tissue repair during diseases or ER stress [43,44].

In the process of collagen biosynthesis, after the export of procollagen molecules to the extracellular matrix, procollagen is cleaved by matrix metalloproteinases (MMPs), generating tropocollagen molecules (trimeric collagen molecules), which self-assemble laterally and longitudinally to form fibrils [40]. Lysine and hydroxylysine residues in trimeric collagen molecules are post-translationally modified to form immature cross-links [12]. The resulting aldehydes then spontaneously react with neighboring lysine residues or other aldehyde residues to generate intra- and intermolecular covalent cross-links, leading to the formation of immature covalent connections, which are converted into stable and mature cross-links that generate macroscopic fibers [41]. These fibers are observed as a part of the normal collagen supramolecular structures in tissues [36] (Figure 1C–E). It is worth mentioning that the use of hydroxylysine and copper sulfate has been established to enhance the formation of collagen cross-links, specifically pyridinoline [42]. This augmentation significantly improves the biomechanical properties of neocartilage [42].

In the fibrillogenesis process, the enzyme-mediated inter- and intramolecular cross-links stabilize tropocollagen molecules within the fibril, which involves lysine, hydroxylysine and histidine residues [7]. Then, covalent cross-links reinforce the mechanical properties of the collagen supramolecular structure in a tissue-specific manner, which is why they are the most crucial post-translational modification described in detail in these revisions [43,44,45]. In addition, this review focuses on enzymatic and non-enzymatic covalent cross-links of type I collagen because, at present, the knowledge of collagen cross-linking has mostly been explored from studies of fibrillar type I collagen.

## 3. Enzymatic Collagen Cross-Linking Mediated by Lysyl Oxidases

It has been well-established that lysyl oxidase acts upon its substrate’s collagen, elastin and nonpeptidyl amines, following a ping-pong mechanism [45,46,47]. For collagens, after the secretion of procollagen molecules outside of the cell, lysyl oxidase catalyzes the conversion of lysine and hydroxylysine residues into aldehydes in nonhelical telopeptide regions. Aldehydes react with lysine or hydroxylysine residues contained in the triple-helical domain on an adjacent collagen molecule to form immature divalent cross-links that can also spontaneously react with another divalent cross-links to create mature trivalent cross-links [47]. Furthermore, it has been observed that when the lysine-derived cross-links are formed by hydroxylysine-derived aldehydes, they are more stable than those formed from the lysine aldehyde pathway [5].

LOX enzymes regulate several biological processes, including extracellular matrix stabilization, cellular growth, and homeostasis [48,49]. Still, the primary role of this enzyme family is to participate in the remodelling of extracellular matrices through the formation of inter- and intrachain cross-links in collagen and elastin. Primarily, LOX enzymes promote the first step in the formation of covalent cross-linking to stabilize collagen fibrils [12,50,51]. Specifically, for fibrillar type I collagen, it has been reported that LOX and LOXL2 can form covalent cross-links in the molecule [52]. It has also been demonstrated that LOX, LOXL2 and LOXL4 cross-link collagen IV, along with indications that LOXL4 cross-links IV via an increase in collagen IV deposition in vascular matrix remodeling [47,53,54,55].

It has been shown that LOX activity is vital for wound-healing but also involved in the pathogenesis of fibrotic diseases; more precisely, its dysregulated activity promotes ineffective or excessive collagen cross-linking, which drives multiple diseases [49,56]. In this context, the formation of collagen cross-linking has been associated with many chronic diseases, including diabetes, cancer metastasis, osteoarthritis and vascular and fibrotic diseases [57,58,59]. In addition, the augmented cross-linking activity of these enzymes is responsible for large insoluble extracellular proteins that are resistant to proteolysis reported in several pathological conditions. It has also been associated with the increased deposition of fibrillar collagens in fibrotic areas. The dysregulation of expression and activity of lysyl oxidases have been found to correlate with numerous diseases and adverse physiological states, including fibrosis in different organs such as the liver, lung, and kidney [60,61,62]. Very recently, it has been proposed that LOXL4 in the main LOX activity and is a critical determinant of collagen cross-linking in lung fibrosis [63]. Among all lysyl oxidases enzymes, the isoforms LOX and LOXL2 are widely associated with metastasis progression because they are needed in the production of a permissive niche to maintain metastatic tumor cell growth [64]. Lastly, it has been proposed that defining tissue-specific variance in collagen cross-linking may help to create biomarkers of pathological connective tissues [13]. 

Five members constitute the mammal LOX, which are classified according to primary structure and functions: LOX, LOXL1, LOXL2, LOXL3 and LOXL4 [12]. Members were divided into two subfamilies based on a phylogenetic study described by the Rodriguez-Pascual group [65]. The first subfamily includes LOX and LOXL1, and the second comprises LOXL2, LOXL3 and LOXL4. These enzymes are classified as copper amine oxidases and display a conserved catalytic domain that contains the copper-binding site [47]. The enzymatic reaction of lysyl oxidases isoforms requires, besides copper, the organic quinone cofactor named lysyl tyrosylquinone (LTQ) (Figure 2A) [66,67]. The copper ion is incorporated into LOX in the trans-Golgi network by ATP7A, a copper-transporting P-type ATPase 1 [47,68]. Three histidine residues, H292, H294 and H296, coordinate the essential copper cation in LOX [47,69]. In LOXL2, the equivalent histidine residues H662, H628, and H630 form the copper-binding site [70,71]. The LTQ is formed by specific residues within the nascent enzyme, derived from tyrosine Y355 and lysine K320 in the LOX isoform. For LOXL2, Mure and colleagues described the spatial arrangement of LTQ between Y689 and K653 and their position relative to the coordination site of Cu^2+^ [66,70,72]. The N-terminal domain differs to the highly conserved catalytic domain: AlphaFold provides a protein structure of hLOX (Uniprot-ID: P28300) [73,74] (Figure 2B,C). The catalytic domain is modelled with quality attributes of confidence and very high confidence according to AlphaFold’s confidence score (blue-colored regions in the protein structure). Only the area around K320 shows low confidence (white-colored region in the protein structure). This is likely due to the flexibility of this region, which allows for conformational adjustments during LTQ co-factor formation, as proposed by Meier and colleagues [70].

On the other hand, the N-terminal domain of the LOX protein differs from the highly conserved catalytic domain. LOX and LOXL1 contain a propeptide region in their N-terminal part, whereas LOXL2, LOXL3 and LOXL4 contain four scavenger receptor cysteine-rich domains (SRCR) [12]. This type of domain has been involved in the proteolytic processing of the LOXL2 isoform by the proprotein convertase PACE4 [47,75]. Two N-glycosylation sites have been described in LOXL2 that are present in the second and fourth SRCR domains [12,76]. There are important differences in the modulation of catalytic activity of LOX isoforms and their molecular mechanisms [75].

It has been proposed that the formation of immature enzymatic cross-links, generated during collagen fibrillogenesis by the enzymatic activity of lysyl oxidase, is a beneficial process in development [2]. In contrast, the formation of mature cross-links damages connective tissues over time and is particularly associated with ageing [39]. Examples of immature reducible and divalent cross-links are the aldimine and keto-imine bonds that form in newly synthesized collagens [7,41]. In particular, aldimine bonds are formed between an aldehyde and an amine group through a condensation reaction, where the carbonyl group of the aldehyde reacts with the amine group to form a Schiff base [1,77]. The resulting molecule is named an aldimine- or imine-containing cross-link (Table 1). It has been established that dehydro-lysinonorleucine (deH-LNL), dehydro-hydroxylysinonorleucine (deH-HLNL) and dehydro-dihydroxylysinonorleucine (deH-DHLNL) are immature divalent cross-links that have shown to be crucial to the formation of more complex cross-links in type I collagen. Ketoimine cross-links are another type of immature divalent cross-links when lysine-keto-norleucine (LKNL) and hydroxylysine-keto-norleucine (HLKNL) are produced via Amadori rearrangements using deH-HLNL and deH-DHLNL cross-links [1] (Table 1). These bifunctional reducible cross-links undergo spontaneous maturation into nonreducible trivalent cross-links, such as pyridinoline and deoxypyridinoline (found in bone and cartilage), pyrrole cross-links (present in bone), arginoline (found in cartilage), and histidinohydroxylysinonorleucine (found in skin) [78,79]. The presence of these specific cross-links highlights their tissue specificity [7]. Interestingly, cross-links associated with histidine were reported to be artefacts found in mass spectrometry [80]. However, this type of cross-linking has been detected in vivo, suggesting that it can be susceptible to the low pH that produces cross-link degradation [1].

When compared to the effects of cross-linking elastin, the repercussions of collagen cross-linking are very different. This is primarily the result of the precise packing of the collagen polypeptide chains into a rigid triple-helix and the self-assembly of these molecules into fibrils, which limits the number of residues that are accessible to lysyl oxidase [81]. Secondly, the presence of hydroxylysine in collagen modifies the subsequent reactions of the initial cross-links. These two factors work in tandem to produce this effect. The lysine or hydroxylysine residue that is present in the short non-helical N- and C-terminal portions of the molecule is oxidized after the lysyl oxidase binds to the freshly formed fibril and oxidizes it. Because of the end-overlap packing, the helical portions of one molecule, which is opposite the nonhelical terminal region of a neighboring molecule, are where the enzyme attaches itself to the fibril [7]. The enzyme does not act on the individual molecules; it only acts on the fibril.

Proteins with a collagen-like domain include complement C1q, mannose-binding protein C, pulmonary-surfactant-associated proteins A1, A2, and D, and gliomedin. Some of them are known as soluble defensive collagens because they have a recognition domain that is contiguous with a collagen-like triple-helical domain [7,82], while gliomedin is known as a membrane collagen [83]. Due to the lack of hydroxylysine residues and the structural constraint for lysyl oxidases to build such links, which require action on the produced fibril, it is likely that such cross-linking does not occur in collagen-like proteins.

Significantly, aging often manifests in two contrasting scenarios: an excessive local deposition of collagen, as seen in fibrosis, or a gradual overall reduction in collagen mass [84]. As individuals age, the normal cross-linking of collagen in connective tissues diminishes due to the cumulative damage from collagen fragmentation, oxidation, and glycation [85]. This progressive decline in collagen mass, observed not only in supporting tissues but also in other organs, compromises the integrity of the extracellular matrix and has implications for age-related conditions like diabetes, cancer, chronic liver disease, and cardiovascular diseases [86]. Simultaneously, the accumulation of molecular damage, chronic inflammation, or injury during aging can drive abnormal collagen deposition, leading to fibrosis [37,87]. Furthermore, the deficiency of the LOX enzyme in adult skin has been associated with inadequate or abnormal collagen cross-linking, which contributes to skin aging [88]. These observations indicate a general decline in enzymatic collagen cross-linking with age. However, due to the increased occurrences of damage or injury, atypical collagen accumulation, as observed in fibrosis, can transpire.

## 4. Lysine-Derived Cross-Links and ECM Proteins

The stable supramolecular structures formed by the cross-linking of collagen molecules range from dense fibers to thin flexible membranes. The formation of these cross-links must be meticulously regulated, as excessive or insufficient cross-linking may result in tissue dysfunction during the organism’s growth [81]. Tissue integrity is maintained through cross-linking, which has been demonstrated for the fibril-forming collagens (types I, II, III, V, and XI) and the fibril-associated collagen (type IX) [89]. Collagen molecules of the same type or of distinct types (I/II, I/III, I/V, II/III, II/IX, II/XI, and V/XI) undergo lysyl-mediated cross-linking at the intramolecular and intermolecular levels [7,78,89]. Collagen IV is stabilized by LOXL2 in glomerular basement membrane [53]. Type VI collagen is one of the filamentous collagens [81]. Type VI, unlike types I through V, appears to be stabilized not by lysyl-aldehyde cross-links but by disulphide bonds [81].

Until now, in this revision, lysyl oxidases were discussed in relation to type I collagen cross-linking in the extracellular matrix (ECM), with an emphasis on their function in ECM construction and remodeling; however, LOX-mediated collagen cross-linking is regulated by extracellular matrix (ECM) proteins (fibronectin, fibulin-4, and thrombospondin-1) and proteoglycans (fibromodulin and syndecan-4) [47]. Fibulin-4, thrombospondin-1, and fibronectin have all been hypothesized to inhibit the proteolytic activation of proLOX, with thrombospondin-1 being able to bind the helical cross-linking sites of collagens. Crosslinks are strengthened by proteins like fibromodulin and syndecan-4 [47]. The degree to which fibrillar collagen is cross-linked is under the control of a small leucine-rich proteoglycan called fibromodulin [90]. Fibromodulin forms links with collagen fibrils as well as LOX, which, in turn, directs the enzyme to N-telopeptide cross-linking sites in collagen I and II fibrils [91]. Syndecan-4 is a membrane proteoglycan that interacts with collagen and stimulates the synthesis of collagen fibers through the use of its extracellular domain [92]. This interaction may help in LOX-mediated collagen cross-linking. Additionally, the contact between periostin and BMP-1 encourages the proteolytic activation of pro-LOX, and it is possible that periostin acts as a scaffold for the interaction of proLOX and BMP-1 on a fibronectin matrix [47]. The enzymes LOX, LOXL1, and LOXL2 cross-link tropoelastin [93]. The main bifunctional cross-links of elastin are dehydrolysinonorleucine and allysine aldol [47]. LOXL2 initiates the formation of dehydrolysinonorleucine and desmosine in elastin. Principal tetrafunctional cross-links of elastin are desmosine or isodesmosine [94]. In addition, collagens and elastin are examples of insoluble extracellular matrix substrates that LOX works upon; however, LOX also acts upon soluble substrates such as TGF and FGF-2, blocking their downstream signaling [47,95,96].

## 5. Lysine-Derived Cross-Link Analysis Methods

Standard procedures have been used to isolate and purify collagen and elastin cross-links derived from lysine [1]. The chemical reduction of imine-based cross-links with sodium borohydride (NaBH4) to produce amine derivates, which increase their stability and prevent extraction procedure susceptibility, has been used to isolate and purify lysine-derived cross-links [79,89]. Despite the fact that mass spectrometry and NMR spectroscopy have predicted their structures, the laborious extraction processes make these cross-links susceptible to degradation. Two divalent nascent cross-links, aldimine and ketoamines, have been shown to be acid- and heat-labile [4]. In addition, pyrrole-containing cross-links (lysyl pyrrole and hydroxylysyl pyrrole) remain difficult to evaluate, primarily because the pyrrole component is sensitive to alkaline or acidic extractions. The original structures of deoxypyrrololine and pyrololine were found by using biotinylated Ehrlich reagents (p-dimethylaminobenzaldehyde) on collagen peptides [1,97]. As another form of lysine-derived cross-link, fluorescent cross-links, such as pyridinium-salt-containing cross-links (pyridinoline and deoxypyridinoline), can be obtained and purified via RP-HPLC with fluorescence detectors. Immunoassays can quantify pyridinolines to monitor osteoporosis, metabolic bone diseases, and osteogenesis imperfecta [4]. Using Fourier transform infrared spectroscopy, pyridinoline and deoxypyridinoline cross-linked peptides, as well as articular cartilage histopathology, have been detected.

Current methods for lysyl- and hydroxylysyl-derived crosslink identification and characterization include ultraperformance liquid chromatography (UPLC), liquid chromatography coupled with mass spectrometry (LC-MS/MS), and immunological techniques. Typically, protein electrophoresis on polyacrylamide gels (SDS-PAGE) and two-dimensional gel electrophoresis are used in combination with MS analysis to determine the presence of cross-linked species in samples [98]. Additionally, the aldehydes that LOXL2 and LOXL3 produce in collagen were located using an aldehyde detection system. We also used Western blotting with carbonyl detection and the oxyblot method to examine protein crosslinks [53]. The creation of a hydrazone derivative of DNPH, which revealed carbonyl groups, was detected using the conventional method of detecting protein carbonyls, Brady’s reagent, 2,4-dinitrophenylhydrazine (DNPH), in combination with immunoblotting performed with an anti-DNPH antibody. The carbonyl groups formed during the LOXL2-mediated cross-linking of 7S subunits were readily detectable using DNPH in our in vitro cross-linking experiments. Although the findings do not provide insight into the precise chemical structure of such cross-links, they are consistent with the analysis of collagen IV from human placenta by Bailey et al. and suggest that DNPH is reacting with newly produced ketoimine groups of hydroxylysine-derived divalent cross-links [53].

Yamaguchi et al., 2021 evaluated the lysyl oxidase-like activity of several polyphenols using ESI-LC-MS/MS and a lysine analog (Bt-APA) and discovered that certain polyphenols, including piceatannol, catechin, and epicatechin, were responsible for mediating the oxidative deamination by monitoring the aldehyde-2-piperidinol intermediates, derivatives of polyphenols [20]. Initial transformation of the lysine derivative to aldehyde is followed by equilibration with 2-piperidinol. The aldehyde reacts with another aldehyde molecule to form aldol. It was determined to be a cross-linking product generated by the aldol condensation reaction between two aldehyde species known to be the source of additional crosslinks in elastin, including desmosine and its isomer isodesmosine. However, no attempt was made to detect these pyridyl crosslinks in this study [20].

Finally, it has been suggested that the chemical synthesis of pure crosslinks, both enzymatic and non-enzymatic, provides a method for producing sufficient material with an overall yield of between 25 and 30 percent for the preparation of an internal standard for the identification of lysyl- and hydroxylysyl-derived crosslinks [1].

## 6. Non-Enzymatic Collagen Glycation

Reducing sugars such as glucose, fructose, pentoses, galactose, mannose and xylulose possess a reactive carbonyl moiety that reacts non-enzymatically, predominantly with the ε-amino group of lysine and the guanidine group of arginine residues, to form a labile compound [1,87,99,100,101]. Then, the early stage in the reaction is the formation of a Schiff base between glucose and protein amino groups, followed by an Amadori rearrangement, a more stable early glycation product [15]. These Amadori products can produce a variety of reactive dicarbonyl compounds such as methylglyoxal and glyoxal, an intermediate stage that can react with other free amino groups of proteins [102,103]. Afterwards, spontaneous reactions occur in the late stage to produce a chemically related group of moieties, termed advanced glycation end products (AGEs), which remain irreversibly bound to proteins. AGE products are a series of spontaneous post-translational modifications found on long-lived proteins, including collagens, and their tissue accumulation is markedly associated with loss of structural integrity in ageing [1,13,37,77,102,104,105,106,107]. It has been shown that protein glycation disrupts normal activities such as enzymatic activity, molecular conformation, degradation ability and receptor recognition [108].

The evidence suggests that AGEs are involved in inflammation and contribute to diverse pathologies, such as cardiovascular and renal diseases, diabetes and many cancer types that are extensively reviewed in [15,100,105,109]. For example, the glycation of collagen by AGE products generated through persistently elevated glucose levels has been suggested to promote the development of fibrosis produced by tissue stiffening and reduce its solubility in diabetes, a chronic disease with hyperglycemia. Various types of AGEs have been studied, and the most common is pentosidine, which exhibits fluorescent properties. Other types of AGEs are the vesperlysine-type AGEs, glyoxal lysine-dimer, methylglyoxal lysine-dimer and glucosepane (Table 1). It has been demonstrated that glucosepane is elevated in elderly patients and diabetic patients [110]. Also, an amide-containing AGE crosslink called GOLA has been correlated with collagen stiffness in rat tendons. A full review was published by Gaar and colleagues, focusing on the enzymatic and non-enzymatic collagen cross-links found in collagen [1].

It has been suggested that the number of lysyl-derived cross-links occurs in significant abundance in comparison with AGEs collagen cross-links in younger persons. However, during the ageing process, collagen glycation increases while lysyl oxidase-mediated cross-links are maintained, which promotes an imbalance in the formation of cross-links [111]. Collagen glycation and the formation of AGE products can alter collagen fibrils’ elasticity, thermal denaturation and morphology (Figure 3) [111]. It has been demonstrated that the target of glycation in type I collagen is the same site that uses lysyl oxidase to generate immature cross-links. Moreover, it has been shown that the glycation on lysine residues in type I collagen has the potential to block normal lysyl oxidase-mediated collagen cross-linking [14]. In fact, lysine positions are more conserved than glycine, hydroxylysine or lysine residues in collagen, and the glycation in this position leads to the loss of enzymatic collagen cross-links [107]. It has been proposed that this effect is mainly because both share the same target in the helical domain, producing a steric hindrance by glucose carbonyls that significantly impairs the normal lysyl oxidase-mediated collagen cross-linking [1,9,14,79]. However, the authors propose the AGE product glucosepane in diabetes and quantified it as the most abundant and relevant AGE product associated with type I collagen molecules [106,110]. By combining experimental and computational analysis, it has been shown that glucosepane alters the density of collagen-packing and the denaturation temperature and enhances the porosity of water molecules in aged tendons [106,111]. Also, high glucosepane levels have been identified in diabetic skin samples and associated with microvascular disease in other organs (Table 1) [110,112]. Until recently, the focus of collagen modification by AGEs and their consequences has mainly been explored with glucose because it is the primary glycating sugar in ageing and diabetes. However, it has been shown that 5-P-Ribose (5PR) has the propensity to glycate collagen, which can modify the cancer microenvironment. A study by Bandose and colleagues describing the effect of ribose on the molecular organization of collagen fibrils found that the impact of R5P glycation on the enzymatic cross-links and molecular arrangement into collagen fibrils, inhibits or breaks the formation of lysyl derived cross-links. This does not only on the surface of the fibrils, because LC-MS experiments showed that R5P glycation is able to reduce the number of HLNL cross-links impacting the fibril structure more than the surface of the collagen molecules, which has significant effects on the arrangement of protomers in the formation of collagen fibrils [107]. It has been proposed that this effect on the changes in the molecular alignment of collagen fibrils can impact cell adhesion and migration processes [107].

## 7. Lysyl Oxidase Activity of Polyphenols

Polyphenols, which are derived from plants, fruits, vegetables, floral tissues, stems, bark, and roots, serve as major sources of polyphenolic compounds, and are the most varied group of phytochemicals [23]. Extensive research has been conducted on these natural products due to their well-known health benefits and protective effects [16,23]. Among the polyphenols, flavonoids play a crucial role and are associated with various positive health outcomes, including increased longevity and reduced risk of cardiovascular diseases in populations with a diet rich in flavonoids [24,113]. These beneficial effects are primarily attributed to the potent antioxidant activity exhibited by polyphenols. However, it is important to note that, under certain conditions, polyphenols can also act as prooxidants, promoting the oxidation of other compounds [20,21,22]. This prooxidant activity may be linked to the non-enzymatic, metal-catalyzed oxidation of polyphenols, leading to the generation of hydrogen peroxide (H_2_O_2_). This non-enzymatic system can lead oxidative modification of proteins by oxidative deamination of the ε-amine group of lysine to α-aminoadipic-5-semialdehyde, the main carbonyl product, similar to the reaction of LOX enzymes [20,21]. Therefore, the biological activities of polyphenols, including their behavior as antioxidants or prooxidants, are believed to be concentration-dependent and are directly proportional to the total number of hydroxyl groups, especially those present in the B-ring of flavonoid molecules [24].

Several studies have described the properties of specific flavonoids in preserving collagen stability [114,115]. For instance, anthocyanidins, natural plant pigments found in fruits, flowers, and certain vegetables, have been shown to stabilize collagens [116,117]. Extracts rich in anthocyanidins inhibit collagen degradation, reduce metalloprotease (MMP) activity, and protect against UV radiation in dermal fibroblast models, thus preventing UV-induced skin photoaging [118]. Furthermore, flavonoids can stimulate the production of fibrillar collagen in mouse fibroblast models. Another mechanism associated with collagen stability preservation involves the inhibition of collagenase and elastase activity [119].

Importantly, polyphenols and flavonoids have been found to contribute to the formation of non-covalent and covalent collagen cross-links [1,114]. It has been established that polyphenols possess amine oxidase-like activity in the presence of Cu^2+^, which is essential for facilitating appropriate collagen and elastin cross-linking [20]. Several polyphenols from different plant species, including chlorogenic acid, gallic acid and caffeic acid, have been associated with this amine oxidase-like activity [21]. It has been assumed that these polyphenols are converted to the o-quinones and acquire a lysyl oxidase-like activity [21,24]. In addition, the conversion of catechol-type polyphenols into o-quinone derivatives has also been observed [20]. These quinones can catalyze the oxidative deamination of primary amines by polyphenols, leading to the formation of iminoquinone and iminophenol, and ultimately resulting in the oxidation product α-aminoadipic-5-semialdehyde [20]. Notably, the catechol-type polyphenols, which includes catechin (C), epicatechin (EC), epigallocatechin (EGC), epigallocatechin gallate (EGCG), gallocatechin gallate (GCG), and epicatechin gallate (ECG), are flavonoids primarily found in green tea and grapes [16,22] (Figure 4). In particular EGCG has shown a high capacity to oxidize specific lysine residues, further contributing to oxidative deamination [20]. The analog of resveratrol, piceatannol, has been associated with the formation of dehydrolysinonorleucine (HLNL) [120].

It has been established that the failure to form covalent cross-links results in the increased degradation of collagen molecules [40,121]. Considering the impact of both polyphenol activities on collagen cross-linking, it is interesting to explore their influence on the regulation of fibrotic processes and wound healing. However, further studies are necessary to elucidate the effect of natural extracts on collagen cross-linking. Most scientific investigations have primarily focused on evaluating the biosynthesis and deposition of the extracellular matrix. Therefore, it is crucial to examine the levels of cross-linking and the molecular mechanisms involved in the interaction between flavonoids and lysyl oxidase activity. This will enable us to establish appropriate methods for accurately measuring collagen cross-linking. Additional research is essential to enhance our understanding of the previously undefined roles of natural extracts and their beneficial effects on collagen biosynthesis and enzymatic cross-linking.

## 8. Anti-Glycating Activity of Polyphenols

At present, a compelling body of evidence supports the anti-glycation activities of polyphenols. Glycation is a spontaneous and non-enzymatic reaction between reducing sugars, such as glucose and fructose, and free amino groups of proteins, DNA, and lipids, which render an unstable Schiff base and then convert it to more stable structures known as Amadori products. These products may undergo a complex series of reactions, leading to the formation of advanced glycation end-products (AGEs). The formation of AGEs occurs at a very high rate in the presence of hyperglycemia and tissue oxidative stress [102].

The pathological implications of AGEs formation are extensively supported in many human diseases. Although initially ascribed to the mechanisms underlying the micro- and macrovascular complications in Diabetes Mellitus [105,122], the contribution of advanced glycation end-products to many human diseases is well-documented [101,123]. The pathologic effects of AGEs are mainly related to their ability to promote oxidative stress and inflammation by binding to the receptor for advanced glycation end-products (RAGE) [124], or by cross-linking with proteins and thus altering their structure and function [125]. Crosslinking by advanced glycation end products increases the stiffness of the collagen network in human articular cartilage: a possible mechanism through which age is a risk factor for osteoarthritis.

Polyphenols can significantly reduce the unhealthy consequences of advanced glycation end-products by different mechanisms. These effects are achieved by interfering with either RAGE expression and signaling, or by inhibiting the cross-linking with body proteins [28]. Notably, the trapping capacity of dicarbonyls compounds, particularly methylglyoxal (MGO) or glyoxal (GO), has been reported for some polyphenols, and deserves special attention because these dicarbonyls are considered one of the most efficient protein crosslinkers [103].

In this regard, several polyphenols displayed important dicarbonyls-trapping activities, as reported for quercetin [126], chrysin derivatives [127], genistein [128], epigallocatechin-3-gallate [129], as well as for resveratrol and different hydroxycinnamic acids [130,131] among many other polyphenols. Of note, this activity has been reported not only for soluble polyphenols but also for bound-polyphenol-rich insoluble dietary fiber [132].

## 9. Conclusions

Collagen, elastin, and other extracellular matrix (ECM) proteins play a crucial role in maintaining tissue integrity, providing structural support, and facilitating cellular functions [121]. The proper regulation of their synthesis, assembly, and cross-linking is essential for maintaining tissue homeostasis [133]. In recent years, the synergistic action of polyphenols and lysyl oxidase (LOX) enzymes has emerged as a fascinating area of research, offering new insights into the modulation of ECM proteins. This essay explores the collaborative relationship between polyphenols and LOX enzymes and their impact on the homeostasis of collagen, elastin, and other ECM proteins. Polyphenols are a diverse group of natural compounds, found abundantly in fruits, vegetables, and plant-based products [23]. They possess antioxidant, anti-inflammatory, and anti-glycation properties, making them potential candidates for maintaining ECM homeostasis. By exerting their antioxidative effects, polyphenols can mitigate oxidative stress-induced damage to ECM proteins, preserving their structural integrity.

LOX enzymes are key regulators of ECM cross-linking, particularly in collagen and elastin fibers [9]. These enzymes catalyze the oxidative deamination of lysine and hydroxylysine residues, leading to the formation of aldehydes that subsequently undergo spontaneous condensation reactions, resulting in the generation of covalent cross-links. LOX enzymes contribute to the maturation and stabilization of ECM proteins, imparting strength and resilience to tissues. In fact, a lysyl oxidase-like activity, resembling an amino oxidase-like activity, has been observed in a various of polyphenols, particularly those of the o-diphenolic type, in the presence of Cu^2+^ ions [20]. However, most studies on the oxidation of lysine residues mediated by polyphenols were conducted using small molecules such as lysine analogs [20]. The potential of these polyphenols to modulate the levels of cross-linking (both immature and mature cross-links) of human type I collagen remains unclear. Further research is required to fully comprehend the impact of polyphenols-mediated cross-linking on the structure and function of collagen.

Emerging evidence suggests a synergistic interaction between polyphenols and LOX enzymes in the maintenance of ECM homeostasis. Polyphenols have been shown to influence LOX enzyme activity and expression levels, thereby impacting the cross-linking process [134]. Certain polyphenols have demonstrated the ability to enhance LOX enzyme activity, promoting the formation of mature and functional cross-links in collagen and elastin fibers [27]. Additionally, polyphenols can protect LOX enzymes from degradation, ensuring their sustained activity and promoting ECM integrity. Beyond their influence on LOX enzymes, polyphenols can directly modulate ECM remodeling processes. These compounds have been shown to regulate matrix metalloproteinases (MMPs) and tissue inhibitors of metalloproteinases (TIMPs), which are involved in ECM turnover and remodeling [119]. Polyphenols can inhibit excessive MMP activity and promote the synthesis of TIMPs, thereby maintaining the balance between ECM synthesis and degradation [40]. Therefore, the interplay between polyphenols and LOX enzymes represents a fascinating area of research, with significant implications for maintaining ECM homeostasis. Through their antioxidant properties, polyphenols protect ECM proteins from oxidative damage, while their ability to modulate LOX enzyme activity and expression influences the formation of cross-links that are critical for collagen and elastin fiber integrity. Furthermore, polyphenols can directly regulate ECM remodeling processes, further contributing to the overall balance between ECM synthesis and degradation. Understanding the synergistic action of polyphenols and LOX enzymes offers exciting prospects for the development of therapeutic strategies aiming to preserveg tissue integrity and addressing ECM-related disorders. In addition, the development of active compounds capable of modulating the oxidative deamination properties of specific natural polyphenols and facilitating the formation of immature and mature cross-links could contribute to the stabilization of collagen macrostructures for various applications in tissue engineering and regenerative medicine, including wound healing. By modulating collagen with polyphenols, the mechanical properties of collagen-based materials can be enhanced, making them more resilient to damage and degradation. Further investigations are warranted to unravel the intricate mechanisms underlying this collaborative relationship and its full therapeutic potential.

Furthermore, the effect of catechol-type polyphenols on the generation of collagen cross-links through enzymatic modifications (LOX and LOXL2) and non-enzymatic (AGE-products) has yet to be explored. The inhibitory and disruptive effects of polyphenols on the formation of advanced glycation end-products (AGEs) in collagen have not been thoroughly investigated either. The interference of several polyphenols with the glycation of lysine ε-amino groups within collagen fibrils, leading to significant trapping of dicarbonyls, could potentially reduce the adverse consequences associated with advanced glycation end-products. However, further studies are necessary to elucidate the effects of natural polyphenols-rich extracts on the enzymatic and non-enzymatic collagen cross-linking.

## Figures and Tables

**Figure 1 ijms-24-10985-f001:**
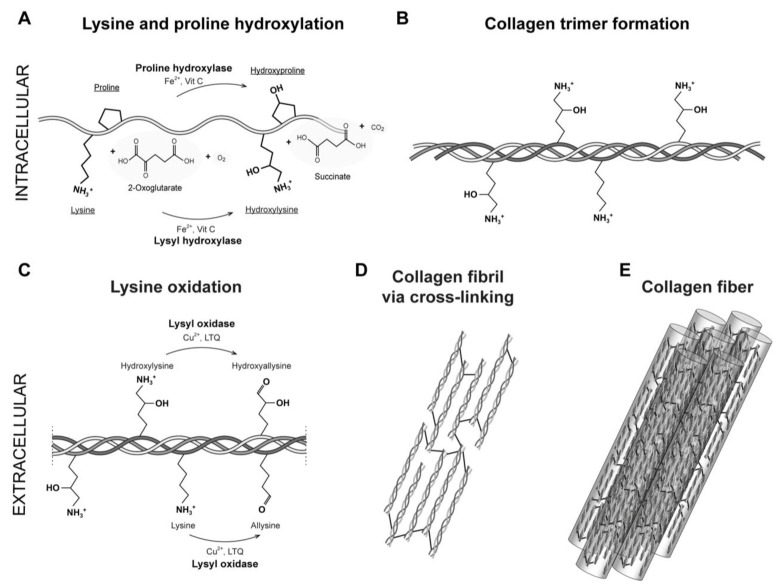
Post-translational modifications involved in the formation of enzymatic collagen covalent cross-linking. Structure of general collagen-forming reaction pathway and its constituent amino acids. Procollagen molecules convert to trimeric propeptide fragments that form tropocollagen molecules. Tropocollagen molecules self-assemble via the reaction of aldehyde groups and the formation of covalent bonds that cross-link collagen molecules into fibrils and fibers. (**A**) The intracellular lysine modification to hydroxylysine by lysyl hydroxylase. Ascorbate and iron act as co-factors for the proline and lysine hydroxylases that stabilize the tertiary structure of collagen molecules. (**B**) The intracellular triple-helix formation highlighting hydroxylysine residues. (**C**) Extracellular lysine oxidation and deamination of lysine to allysine and hydroxylysine to hydroxyallysine. The LOX superfamily modifies lysine and hydroxylysine residues in collagen post-translationally. (**D**,**E**) showing mature cross-linked tropocollagen, which organizes into collagen fibrils and assembles into collagen fibers.

**Figure 2 ijms-24-10985-f002:**
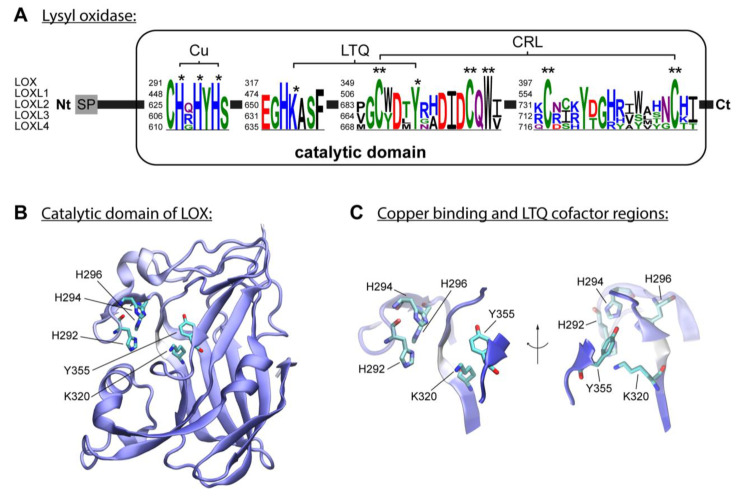
**The lysyl oxidase family**. (**A**) LOX isoforms contain a signal peptide (**SP**) in the amino-terminal region and a catalytic domain in the highly conserved carboxyl-terminal region. The catalytic domain includes a copper-binding domain (**Cu**), lysyl tyrosylquinone cofactor (**LTQ**) and cytokine receptor-like domain (**CRL**), which are also present in isoforms LOXL1, LOXL2, LOXL3 and LOXL4. Three histidine residues forming the copper-binding site are conserved in human LOX isoforms. Respective histidine residues are marked with an asterisk in the sequence logos based on the five human LOX isoforms. The conserved lysine y tyrosine residues forming the LTQ are also marked with an asterisk in the sequence. Conserved residues of the CRL domain are marked with two asterisks. (**B**) AlphaFold model of hLOX lysyl oxidase-like domain represented in NewCartoon. The color illustrates the confidence of the model structure, with white indicating low-confidence regions and blue high-confidence regions. Residues H292, H294 and H296, and LTQ cofactor residues K320 and Y355, which form copper-binding sites, are highlighted. (**C**) Detailed view of the residues highlighted in (**B**). Note that the confidence of the structure around residue K320 is low, which indicates uncertainty in this region (see main text for more information).

**Figure 3 ijms-24-10985-f003:**
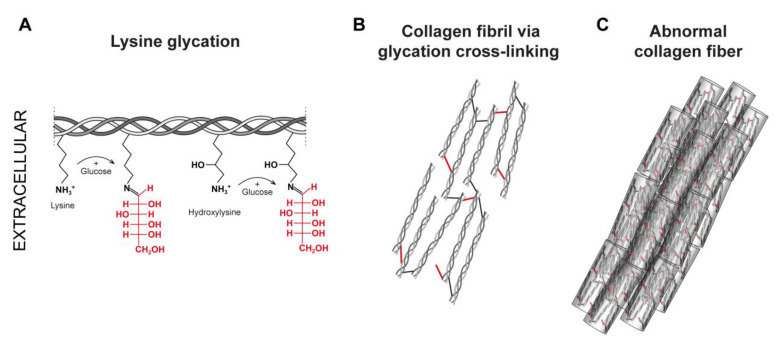
**Glycation of Collagen Fibers**. Glycation of extracellular lysine by glucose to produce advanced glycation end products (AGE). Glycation of some lysine and hydroxylysine, as well as oxidation (collagen glycoxidation), both contribute to the formation of the non-enzymatic cross-links known as AGEs. Certain AGEs have the ability to increase the blocking of LOX cross-linking sites, which results in the fibrils being less stable. Age-associated glycosylation end products (AGEs) have been shown to increase with age, notably in populations with inflammatory diseases and diabetes. (**A**) Reducing glucose has a reactive carbonyl moiety that forms a Schiff base by reacting non-enzymatically with the -amino group of lysine to produce a labile molecule. (**B**,**C**) display mature tropocollagen that has been cross-linked by glycation. Additionally, there is disorganization within the collagen fibrils and the suprafibrillar architecture is seen. Glycation of collagen is shown by the bonds in the color red.

**Figure 4 ijms-24-10985-f004:**
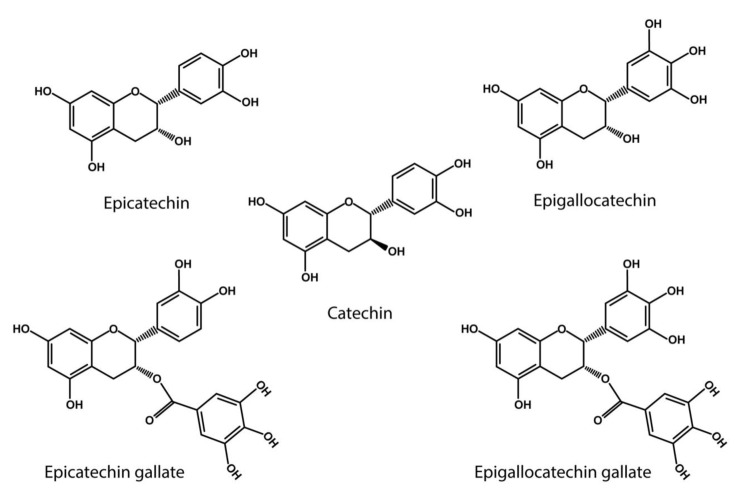
**Catechol- type polyphenol structures**: Chemical structures of catechin derivatives, which are proposed as putative quinone cofactors due to their amine oxidase-like activity, which could be attributed to polyphenols, or where these polyphenols could be required. Catechin (C), epicatechin (EC), epigallocatechin (EGC), epigallocatechin (EGCG), gallocatechin gallate (GCG), epicatechin gallate (ECG).

**Table 1 ijms-24-10985-t001:** Types of covalent collagen cross-links.

Immature Cross-Links	Mature Cross-Links	Glycation-Derived Cross-Links
**Aldimine cross-links** Dehydro-lysinonorleucine (deH-NL) Dehydro-hydroxylysinonorleucine (deH-HLNL) Dehydro-dihydroxylysinonorleucine (deH-DHLNL)	**Pyrrole-containing cross-links** Deoxypyrrololine (d-Prl) Pyrrololine (Prl) **Pyridinium-salt-containing cross-links** Pyridinoline (Pyr) Deoxypyridinoline (Dpyr)	**Alpha-dicarbonyl compounds (α-DC)** Methylglyoxal (MGO) Glyoxal (GO)
**Ketoamine cross-links** Lysine-keto-norleucine (LKNL) Hydroxylysine-keto-norleucine (HLKNL)	**Histidine-containing cross-links** Histidinohydroxylysinonorleucine (HHL) Histidinohydroxymerodesmosine (HHMD) **Arginoline** Ketoimine cross-link derivate	**Fluorescent cross-linking AGEs** Pentosidine (PEN) **Non-Fluorescent cross-linking AGEs** Carboxymethyl-lysine (CML) Glucosepane (GSP)

Immature reducible and divalent cross-links are aldimine and keto-imine bonds. Mature multivalent cross-links include pyrrole-containing cross-links (deoxypyrrololine (d-Prl) and pyrrololine (Prl)), pyridinium-salt containing cross-links (pyridinoline (Pyr) and deoxypyridinoline (Dpyr)) and histidine-containing cross-links (histidinohydroxylysinonorleucine (HHL) and histidinohydroxymerodesmosine (HHMD)). Glycation derived cross-links include dicarbonyl compounds and advanced glycation end products (AGEs).

## Data Availability

Not applicable.

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
