# Peer review of "Exploring the Interplay between Polyphenols and Lysyl Oxidase Enzymes for Maintaining Extracellular Matrix Homeostasis"

_ijms, 2023, doi:10.3390/ijms241310985_

Round 1

Reviewer 1 Report

Añazco et al write an excellent review on collagen biochemistry and shed light on an understudied interaction in collagen literature - the potential role of polyphenols in maintain ECM homeostasis. The authors clearly define each topic, discuss concisely, and gives the reader essential background moving through the review.

Additional section required: The authors should add an additional section regarding N- and O-linked glycosylation. The authors briefly mention these collagen post-translational modifications, but a complete section reviewing what is known regarding the role of enzyme-mediated N- and O-glycosylation is needed. This distinction is especially necessary considering the emphasis of the review on hydroxylysine. Similarly, many scientists frequently confuse glycation and glycosylation, so the field would benefit from focused reviews that clearly outline the distinction in the context of extracellular matrix biology. 

Reviewer 2 Report

The review of Añazco and colleagues provides a comprehensive overview of collagen cross-linking, both non-enzymatic and mediated by lysyl oxidases, and suggests that polyphenols can be therapeutically used to regulate this process, particularly in aging. It is a well-written, informative and stimulating review. The focus is on collagens, although elastin is also mentioned in some instances. Could this aspect be expanded into a separate section, mentioning other ECM molecules that are cross-linked to each other or with collagens via this type of linkage? This would increase the impact of the review. Are there any known or even speculated functions that such cross-linking would have for not collagens? Obviously, proteins having collagen-like domains such as C1q or gliomedin would be of first interest, but would it also be related to lecticans as a very important family of ECM molecules in the cartilage and the brain? What are the structural constraints for lysyl oxidases to form such links?

What are the methods to quantify these cross-links on proteins of interest and acutely cut them in vivo to study the functional impact? Introducing briefly such info may stimulate further research and collaborations in the field.

Minor points:

Line 228: three

Figure 3 is not much different from Figure 1, the differences in fibril structures should be made more visible (disorganized).

The quality of English is satisfactory.

Round 2

Reviewer 1 Report

Thank you for your detailed response to comments. I believe the authors have added a sufficient expansion on previously lacking sections and this manuscript is suitable for publication in it's present state.